# Cognitive function in multiple sclerosis: A long-term look on the bright side

Yermi Harel[1,2,3�], Alon Kalron[4,5�], Shay Menascu[1], David Magalashvili[1], Mark Dolev[1], Glen Doniger[5,6,7], Ely Simon[6†], Anat Achiron[1,3,5]*

**1** Multiple Sclerosis Center, Sheba Medical Center, Tel Hashomer, Israel, **2** Loewenstein Rehabilitation Hospital, Raanana, Israel, **3** Sackler Faculty of Medicine, Tel Aviv University, Israel, **4** Department of Physical Therapy, School of Health Professions, Sackler Faculty of Medicine, Tel-Aviv University, Israel, **5** Sagol Neuroscience Center, Tel-Aviv University, Tel-Aviv, Israel, **6** Department of Clinical Research, NeuroTrax Corporation, Medina, NY, United States of America, **7** Center of Advanced Technologies in Rehabilitation, Sheba Medical Center, Tel Hashomer, Israel

he These authors contributed equally to this work.
† Deceased.
* Anat.Achiron@sheba.health.gov.il

## Abstract

### Background

Multiple sclerosis (MS) may lead to cognitive decline over-time.

### Objectives

Characterize cognitive performance in MS patients with long disease duration treated with disease modifying drugs (DMD) in relation to disability and determine the prevalence of cognitive resilience.

### Methods

Cognitive and functional outcomes were assessed in 1010 DMD-treated MS patients at least 10 years from onset. Cognitive performance was categorized as high, moderate or low, and neurological disability was classified according to the Expanded Disability Status Scale (EDSS) as mild, moderate or severe. Relationship between cognitive performance and disability was examined.

### Results

After a mean disease duration of 19.6 (SD = 7.7) years, low cognitive performance was observed in 23.7% (N = 239), moderate performance in 42.7% (N = 431), and 33.7% (N = 340) had high cognitive performance, meeting the definition of cognitively resilient patients. Within the group of patients with low cognitive performance, severe disability was observed in 50.6% (121/239), while in the group of patients with high cognitive performance, mild disability was observed in 64.4% (219/340). Differences between the group of patients with high cognitive performance and severe disability (4.5%) and the group of patients with low cognitive performance and mild disability (5.0%) were not accounted for by DMD treatment duration.

**Data Availability Statement:** All relevant data are within the paper.

**Funding:** This research was supported by Sheba MS research grant (AA-1171). The funder had no

role in study design, data collection and analysis, decision to publish, or preparation of the manuscript. I confirm that Glen Doniger and Ely Simon are affiliated with NeuroTrax Corporation and that NeuroTrax Corporation provided support in the form of salaries for these authors (GD and ES), but did not have any additional role in the study design, data collection and analysis, decision to publish, or preparation of the manuscript. The specific roles of these authors are presented in the 'author contributions' section.

**Competing interests:** Glen Donigerand Ely Simon are affiliated with NeuroTrax Corporation. There are no patents, products in development or marketed products to declare. This does not alter our adherence to all the PLOSONE policies on sharing data and materials.

## Conclusions

The majority of DMD treated MS patients did not have cognitive decline that could impair their quality of life after disease of extended duration.

## Introduction

Cognitive impairment has been reported to occur in 40–65% of patients with multiple sclerosis (MS) and can present even in early phases of the disease [1–4]. We have previously reported the profile of cognitive decline in a large cohort of 1500 MS patients showing that cognitive performance was below the normative average for cognitively intact individuals of similar age and education, with information processing speed and executive function most frequently impaired [5]. Cognitive impairment was significant only at disease duration greater than five years suggesting the existence of an early therapeutic window [5]. However, the effects of disease modifying drugs (DMD) on cognitive impairments in MS have not been thoroughly studied [6], though several DMD have demonstrated a beneficial effect on cognitive performance [7] and spared brain atrophy [8–10], thus showing the potential to decrease cognitive decline.

Progression of cognitive decline in MS over time is variable, and it is not yet clear why some patients are cognitively resilient, while others decline within a short period of time. Prevalence of cognitive resilience in MS may explain variability across patients in profile of cognitive decline, and such resilience may signify a less active or even benign disease, and/or improved ability to recover during the active disease process.

In the current study, we characterized cognitive performance in DMD treated MS patients with disease duration longer than 10 years. We computed prevalence of patients with high, moderate and low cognitive performance and evaluated the relationship between cognitive status and neurological disability.

## Methods

### Study design and participants

This was a retrospective analysis of cross-sectional data obtained from RRMS and SPMS patients treated and followed at the Multiple Sclerosis Center, Sheba Medical Center, Tel-Hashomer, Israel. Demographic, clinical and cognitive data were extracted from the Sheba MS Center computerized database. The following criteria were applied to extract data for the current analyses: Inclusion criteria: (1) diagnosis of definite MS according to the revised McDonald criteria [11]; (2) cognitive assessment after at least 10 years from disease onset; (3) neurological examination with Expanded Disability Status Scale (EDSS) [12] within 3 months of the cognitive assessment; (4) treatment with DMD for at least 6 moths. Exclusion criteria: (1) primary progressive disease course; (2) corticosteroid treatment up to 3 months prior to the cognitive assessment; (3) known psychiatric illnesses (including major depression or anxiety) or dementia; (4) alcohol or drug abuse; (5) severe impairment of the upper limbs and/or visual impairment precluding performance of the computerized cognitive assessment. This was determined as a part of the cognitive test. Patients with severe upper limb dysfunction like paralysis or tremor that were not able to hold the computer-mouse were technically excluded from preforming the test. Similarly, at the beginning of the cognitive test visual acuity is assessed and patients that could not read the instructions were technically excluded from performing the test.

Each patient record was coded anonymously to ensure confidentiality during statistical analyses. For patients with multiple cognitive assessments, an automated algorithm randomly selected data from one visit so that each patient is represented only once in the study dataset. The study was approved by the Sheba Medical Center IRB Ethics Committee.

### Cognitive assessment

Cognitive performance was assessed with a battery of computerized tests (NeuroTrax Corporation, Medina, NY, USA). The NeuroTrax cognitive battery has been previously validated in MS patients showing good discriminant and construct validity as compared to conventional cognitive assessment [3, 13], and incorporates alternate forms that minimize learning on follow-ups. The battery is easily administered, and testing included the following cognitive domains: memory (verbal and nonverbal), executive function, attention, information processing speed, visual spatial processing, verbal function and motor skills. Each cognitive score was standardized relative to age-/education-stratified cognitively intact norms and scaled to an IQ-style scale (mean: 100, SD: 15). Domain scores were computed as the average scores from particular tests (see [3,5] for more details). A global cognitive score (GCS) was computed as the average of the cognitive domains scores. Testing time was approximately 45 min. The computerized cognitive battery has shown good test-retest reliability and construct validity relative to paper-based tests, including the frequently used Neuropsychological Screening Battery for MS, NSBMS, [3], as well as sensitivity to effects of DMD in MS [7].

### Group categorization

Cognitive performance was categorized by GCS as "high" (GCS >100), "moderate" (GCS 85–100) or "low" (GCS<85). Neurological disability was classified according to EDSS score as "mild" (EDSS ≤3), "moderate" (EDSS 3.5–5.5), or "severe" (EDSS ≥6.0).

### Statistical analysis

Descriptive statistics were used to summarize demographic, clinical and cognitive variables. Between groups differences were evaluated by the chi-square test for categorial variables and by analysis of variance (ANOVA) for continuous variables. Analyses were carried out using SPSS Version 25.0 (IBM Corporation, Armonk, NY, USA). Two-tailed statistics were used, and significance level was set to p<0.05.

## Results

We analyzed data obtained from 1010 relapsing-remitting (RRMS) and secondary-progressive (SPMS) patients, 700 females, 310 males, mean age 49.3 (SD = 11.0) years, all treated with DMD for a mean period of 9.2 (SD = 5.6) years. Demographic, neurologic and cognitive data subdivided by cognitive performance group are shown in Table 1. After a mean disease duration of 19.6 (SD = 7.7) years, low cognitive performance was found in 23.7% (N = 239) of patients, moderate performance in 42.7% (N = 431), and 33.7% (N = 340) had high cognitive performance attaining scores above average for cognitively intact individuals of similar age and education. Generally, lesser cognitive impairment was associated with lesser disability (Fig 1). As expected, significant differences between the low, moderate and high cognitive groups were found for all cognitive measures. Neurological disability by EDSS score and the functional system disability scores (except for visual functional score) were significantly higher in the groups with low and moderate cognitive performance as compared to patients with high cognitive performance.

**Table 1. Demographic and clinical data for MS patients with long disease duration subdivided by cognitive performance.**

| Variable | Cognitive performance | | | | p-value |
|---|---|---|---|---|---|
| | All | High GCS>100 | Moderate 85≤GCS≤85 | Low GCS<85 | |
| Count (%) | 1010 | 340 (33.7) | 431 (42.7) | 239 (23.7) | |
| Age, y | 49.3 (11.0) | 49.2 (11.0) | 49.7 (11.3) | 48.6 (10.4) | 0.423 |
| Gender | | | | | |
| Female, n | 700 | 219 | 318 | 163 | 0.017*[a,b,c] |
| Male, n | 310 | 121 | 113 | 76 | |
| MS Type (RR/SP) | 812/198 | 298/42 | 352/79 | 162/77 | <0.001*[a,c] |
| Disease duration, y | 19.6 (7.7) | 18.6 (6.9) | 19.8 (8.2) | 20.8 (7.8) | 0.113 |
| Education, y | 14.5 (2.4) | 15.0 (2.5) | 14.3 (2.3) | 14.0 (2.4) | <0.001*[a,c] |
| DMD treatment duration, y | 9.2 (5.6) | 9.2 (5.5) | 9.2 (5.6) | 9.1 (5.6) | 0.979 |
| EDSS | 3.8 (2.2) | 2.8 (2.0) | 3.8 (2.2) | 5.1 (2.1) | <0.001*[a,b,c] |
| Pyramidal | 2.3 (1.4) | 1.7 (1.3) | 2.3 (1.4) | 3.0 (1.3) | <0.001*[a,b,c] |
| Cerebellar | 1.3 (1.2) | 0.8 (0.9) | 1.3 (1.1) | 2.0 (1.2) | <0.001*[a,b,c] |
| Brainstem | 0.6 (0.9) | 0.4 (0.7) | 0.5 (0.8) | 1.0 (1.0) | <0.001*[a,c] |
| Sensory | 1.1 (1.2) | 0.8 (1.0) | 1.2 (1.2) | 1.4 (1.2) | <0.001*[a,b,c] |
| Bowel & Bladder | 1.3 (1.2) | 0.9 (1.0) | 1.3 (1.2) | 1.8 (1.3) | <0.001*[a,b,c] |
| Visual | 0.4 (0.9) | 0.4 (0.9) | 0.4 (0.9) | 0.5 (1.0) | 0.245 |
| Cerebral | 0.3 (0.8) | 0.1 (0.4) | 0.3 (0.8) | 0.6 (1.1) | <0.001*[a,b,c] |
| Global cognitive score | 92.8 (14.0) | 106.2 (4.0) | 93.6 (4.3) | 72.4 (9.9) | <0.001*[a,b,c] |
| Memory | 92.9 (17.9) | 105.2 (6.6) | 95.0 (11.8) | 71.6 (19.0) | <0.001*[a,b,c] |
| Executive function | 93.0 (15.8) | 106.8 (8.7) | 92.6 (8.6) | 73.7 (13.3) | <0.001*[a,b,c] |
| Visual spatial | 97.8 (18.9) | 110.6 (1.4) | 96.6 (15.4) | 81.2 (19.8) | <0.001*[a,b,c] |
| Verbal function | 93.6 (20.4) | 104.4 (8.4) | 94.8 (15.4) | 73.3 (27.2) | <0.001*[a,b,c] |
| Attention | 91.9 (17.3) | 104.9 (6.3) | 93.7 (9.2) | 69.4 (17.4) | <0.001*[a,b,c] |
| Processing speed | 93.6 (17.9) | 107.4 (12.0) | 91.3 (12.6) | 71.2 (12.9) | <0.001*[a,b,c] |
| Motor skills | 92.2 (17.2) | 104.2 (8.9) | 91.3 (12.7) | 71.3 (17.7) | <0.001*[a,b,c] |

GCS = Global cognitive score; DMD = Disease modifying drugs

*[a] between high and low cognitive performance

[b] between low and moderate cognitive performance

[c] between high and moderate cognitive performance.

No significant differences between cognitive performance groups were observed for age and DMD treatment duration.

The relationship between cognitive performance and disability status in the entire cohort (Fig 2), further elucidates the association between disability and cognitive decline.

Among patients with high cognitive performance, 64.4% (219/340) had mild disability and can therefore be considered to have a benign MS disease pattern under DMD treatment. Among patients with moderate cognitive performance, 44% (190/431) had mild disability, and among patients with low cognitive performance, 50.6% (121/239) had a high level of disability. Analysis of patients with 'inconsistency' between cognitive performance and disability status demonstrated that 4.5% (46/1010) had high cognitive performance in spite of severe disability, while 5.0% (50/1010) had low cognitive performance in spite of mild disability. The majority of patients with high cognitive performance and severe disability had SPMS and more years of education as compared to patients with low cognitive performance and mild disability, and the

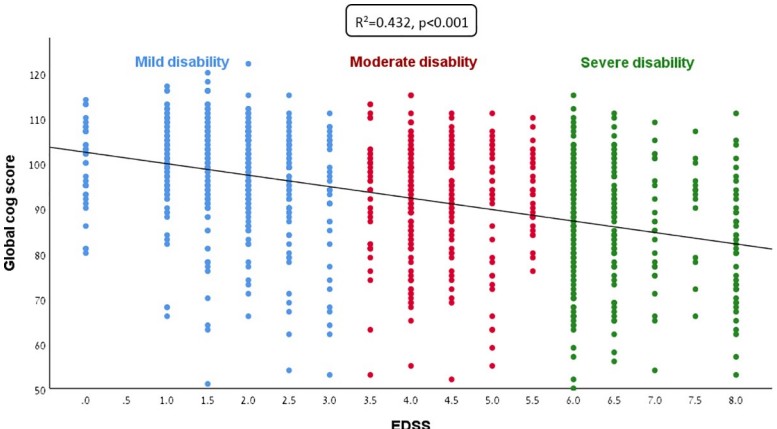

**Fig 1. Correlation between cognitive performance and disability.**

difference between these groups could not be accounted for by DMD treatment duration, Table 2.

## Discussion

Our study assessed the frequency of cognitive impairment in a large cohort of DMD- treated RRMS and SPMS patients after nearly 20 years of illness. Notably, we observed a much lower rate of cognitive impairment than previously reported in the literature. Most studies assessing cognitive performance in MS estimated frequency of cognitive decline, particularly in the domains of attention, processing speed and working memory, to be within the range of 40% to 60% over the lifespan, but these studies were performed mainly in patients who did not receive DMD [4, 14]. It is not surprising that cognitive impairment would be less than in the pre-DMD treatment era, due to effective disease modification and greater ascertainment of milder cases of MS.

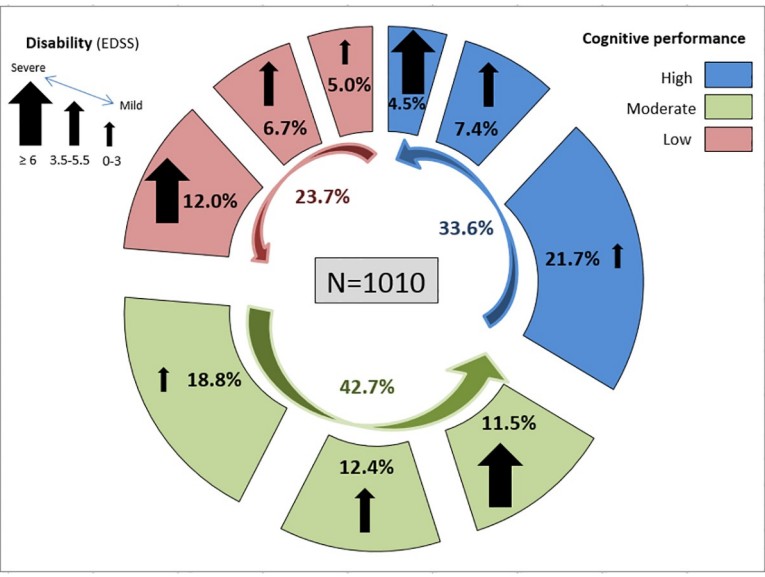

**Fig 2. Cognitive performance in relation to disability levels.**

**Table 2. Groups with 'inconsistency' between cognitive performance and disability.**

| Variable | Low cognitive performance & Mild disability | High cognitive performance & Severe disability | p-value |
|---|---|---|---|
| Number | 50 | 45 | |
| Age, y | 47.2 (10.0) | 56.6 (8.4) | 0.082 |
| Gender (F/M) | 38/12 | 29/16 | 0.191 |
| MS type (RR/SP) | 49/1 | 11/34 | <0.001 |
| Disease duration, y | 19.5 (6.8) | 23.7 (8.1) | 0.789 |
| Education, y | 13.8 (2.2) | 15.6 (2.5) | 0.002 |
| DMD treatment duration, y | 8.7 (5.2) | 10.4 (5.1) | 0.120 |

Data are presented as mean (SD)

The use of DMD significantly changed the pattern of MS progression. These medications modulate the inflammatory immune response and lead to decreased disease activity by reducing relapse rate and delaying disability progression. Consequently, these treatments lead to better cognitive protection [15]. The safety-risk profile of DMD in MS is favorable. Possible side effects vary between treatments and include injection-site reactions, increased risk for infections, gastrointestinal symptoms, flushing, autoimmune thyroid disorders and elevated liver enzymes [16].

In our cohort of patients treated with DMD for a relatively long period, only 23.7% had low cognitive performance, while 33.7% had high cognitive performance attaining scores above average for cognitively intact individuals of similar age and education. This group of patients, that after a long disease duration of almost 20 years, maintained their cognitive skills, can be defined as 'cognitively resilient' patients. The term resilience is derived from the Latin words salire (to leap or jump), and resilire (to spring back). When applied to cognition it denotes the capacity of the brain to resist deteriorating processes or injuries [17,18]. Cognitive resilience literature has focused on specific contexts in which individuals differ in their capabilities to withstand or overcome brain insults and to explain the difference in the patterns of cognitive decline associated with aging and neurodegenerative diseases [19]. Various predisposing interacting factors may contribute to the road map for brain resilience, including education, gender, prior brain injuries, family history, participation in cognitively stimulating activities, physical exercise, social relationships and apoE genotype [20–24]. In young patients with a chronic long-lasting disease like MS, characterizing differences between cognitive subgroups in relation to clinical variables may afford new insights into active neuroplasticity mechanisms and thus suggest plasticity facilitating treatments to enhance cognitive resilience. Our findings suggest a reason for optimism relative to the previously reported studies. Furthermore, we found that approximately 75% of RRMS and SPMS patients (i.e., "moderate" and "high" cognitive performance groups combined) do not have low levels of cognitive function that may compromise quality of life, social interactions, employment prospects and work performance [25].

It is of note that different DMD may have varying effects on cognitive performance, that during the long-term study period, patients switched DMD, and that treatment duration under each DMD varied; however, in spite of these limitations that make it difficult to assess the contribution of each DMD to cognitive performance, our findings are encouraging in suggesting that overall, long-term treatment with DMD affords significant beneficial effects in the maintenance of normal brain function.

As anticipated, increased neurological disability correlated with lower cognitive performance, indicating that cognitive function is an integral clinical feature of MS and directly

related to neurological disability. Neurological disability by the EDSS score and functional system disability scores were significantly higher in the groups with low and moderate cognitive performance as compared to patients with high cognitive performance. No difference was found for visual functional score between the groups, probably because visual impairment was an exclusion criterion. However, no significant differences between high, moderate and low cognitive performance groups were observed for age and DMD treatment duration, suggesting that differences in cognitive performance may be attributable to differential disease activity. More aggressive disease activity characterized by increased rate of relapses and post-relapse residual disability [26] may lead to worse cognitive performance and higher disability despite treatment with DMD; this more aggressive group comprised 12.0% of the cohort. In contrast, 21.7% of the patients had high cognitive performance and low level of disability and can therefore be considered to have benign MS [27, 28]. Cognitive resilience in these patients probably signifies MS resilience, e.g., resilience to the disease pathogenic mechanisms. These patients have low disability and therefore also manifest with less cognitive impairment, suggesting that cognition as a part of the functional neurological spectrum of MS, is better preserved in patients with less active disease. Indeed, it is expected that patients with a benign disease course, probably presenting as good responders to DMD treatment, will be resilient to disability and cognitive decline even after many years of disease.

More intriguing is to understand the groups for which cognitive performance and disability were inconsistent, i.e., patients who after a long-term follow-up present with high cognitive performance and severe disability (4.5%), or patients with low cognitive performance and mild disability (5.0%). We believe that these minority groups may have a different anatomical pattern of central nervous system involvement. Patients with cognitive resilience but severe disability probably have more pronounced spinal cord involvement, either cervical, thoracic or both [29], while patients with low cognitive performance and mild disability likely will have more distinct brain atrophy and a higher brain lesion load, with lesser cervical or thoracic spinal cord involvement [30, 31].

In our cohort, 33.7% of patients were cognitively resilient. These patients had intact cognitive performance despite an extended disease duration, and the majority were also neurologically protected, reflected by a relatively low EDSS score. One of the mechanisms that accounts for cognitive resilience may be preserved brain functional reserve. In accordance, Rocca et al., [32] reported that relatively mild grey matter damage was associated with a favorable clinical course in patients with benign MS. Sumowski et al., [33] have suggested that the level of cerebral efficiency prior to disease affords a 'cognitive reserve' against disease-related cognitive impairment, such that when cognitive processing is challenged by a brain disease, individuals with greater cerebral efficiency are better able to withstand the insult and will not develop cognitive impairment. Indeed, in our cohort, educational level was higher in patients with high cognitive performance as compared to patients with moderate and low cognitive performance.

Although our study is retrospective and cross-sectional and included only relapsing-remitting and secondary progressive patients, it encompasses a large number of patients, and to the best of our knowledge, this is the first study that evaluates "real-world" experience of DMD treatments on cognitive performance in MS patients. We could not evaluate the effects of individual DMD on cognitive performance as patients changed treatments, and the treatment period for each medication varied. However, the main strength of our study is in elucidating the relationship between cognitive performance and disability in MS patients after a long disease duration.

We suggest that pre-disease intellectual enrichment reflected by educational level contributes to better cognitive reserve and better cognitive resilience in MS patients. Furthermore, a cognitive sparing effect can be enhanced by DMD treatment leading to a high rate of MS patients without cognitive decline even after many years from onset.

## Acknowledgments

The authors thank Roi Aloni, Keren Gutman and Tali Paz for their field work that greatly contributed to this research project.

Eli Simon passed away before the submission of the final version of this manuscript. Anat Achiron accepts responsibility for the integrity and validity of the data collected and analyzed.

The authors wish to dedicate this work in memory of our dear co-author Dr. Ely Simon, an expert cognitive neurologist and innovator, who passed away unexpectedly during preparation of this paper. Dr. Simon's warmth, professionalism, and vision have left an indelible mark on the fields of neurology and cognitive assessment.

## Author Contributions

**Conceptualization:** Yermi Harel, Alon Kalron, Anat Achiron.

**Data curation:** Shay Menascu, David Magalashvili, Mark Dolev, Anat Achiron.

**Formal analysis:** Alon Kalron.

**Funding acquisition:** Anat Achiron.

**Investigation:** Yermi Harel, Alon Kalron, Shay Menascu, David Magalashvili, Mark Dolev, Glen Doniger, Ely Simon, Anat Achiron.

**Methodology:** Yermi Harel, Alon Kalron, Glen Doniger, Ely Simon, Anat Achiron.

**Project administration:** Alon Kalron, Anat Achiron.

**Resources:** Alon Kalron, Shay Menascu, David Magalashvili, Mark Dolev, Anat Achiron.

**Software:** Glen Doniger, Ely Simon.

**Supervision:** Yermi Harel, Alon Kalron, Glen Doniger, Anat Achiron.

**Validation:** Yermi Harel, Alon Kalron, Glen Doniger, Ely Simon, Anat Achiron.

**Visualization:** Yermi Harel, Alon Kalron, Glen Doniger, Ely Simon, Anat Achiron.

**Writing – original draft:** Yermi Harel, Alon Kalron, Ely Simon, Anat Achiron.

**Writing – review & editing:** Yermi Harel, Alon Kalron, Shay Menascu, David Magalashvili, Mark Dolev, Glen Doniger, Ely Simon, Anat Achiron.

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
