## [Editor Report · Decision Letter 0]

18 Jun 2019

PONE-D-19-14839

Cognitive function in multiple sclerosis: A long-term look on the bright side

PLOS ONE

Dear Prof ACHIRON,

Thank you for submitting your manuscript to PLOS ONE. After careful consideration, we feel that it has merit but does not fully meet PLOS ONE’s publication criteria as it currently stands. Therefore, we invite you to submit a revised version of the manuscript that addresses the points raised during the review process.

We would appreciate receiving your revised manuscript by Aug 02 2019 11:59PM. To enhance the reproducibility of your results, we recommend that if applicable you deposit your laboratory protocols in protocols.io, where a protocol can be assigned its own identifier (DOI) such that it can be cited independently in the future. For instructions see: http://journals.plos.org/plosone/s/submission-guidelines#loc-laboratory-protocols

We look forward to receiving your revised manuscript.

Kind regards,

Abiodun E. Akinwuntan, PhD, MPH, MBA

Academic Editor

PLOS ONE

Journal Requirements:

Additional Editor Comments:

The manuscript reports on the cognitive performance in individuals with long history of MS treated with DMD in relation to disability and cognitive resilience.

Overall, there needs to be consistencies in terminologies throughout the manuscript. Low cognitive, moderate cognitive, and high cognitive performance or severe cognitive, moderate cognitive, and low cognitive impairments. Specifically, define cognitive resilience in light of the terminology you choose. These will facilitate better comprehension of the manuscript.

Abstract: Apart from the inconsistency in terminologies, this section is well-written.

Introduction: Short and succinct.

Methods: Reads well.

Results: Page 6, end of the first paragraph: As shown in Table 1, there was also no significant difference in visual functions. This fact needs to be included in the narrative as well.

Figure 1 is exactly the same information as in Table 2. Either one, plus the narrative should be sufficient.

Discussion: Page 7, end of the first paragraph: Can you discuss some of side effects of DMDs?

Page 9: beginning of the second paragraph: If this is a prospective study, how is it possible that data was collected prospectively?
---

## [Author Response · Author response to Decision Letter 0]

28 Jun 2019

23.6.2019

Dear Editor

Enclosed please find our revised manuscript PONE-D-19-14839 entitled “Cognitive performance in multiple sclerosis: A Long-term look on the bright side”, which we submit after revision after we addressed the points raised during the review process as follows:

Comment 1. Overall, there needs to be consistencies in terminologies throughout the manuscript. Low cognitive, moderate cognitive, and high cognitive performance or severe cognitive, moderate cognitive, and low cognitive impairments. 

Specifically, define cognitive resilience in light of the terminology you choose. These will facilitate better comprehension of the manuscript.

Response: As suggested, we have better defined the cognitive terminologies throughout the manuscript, using the terminology of low cognitive, moderate cognitive, and high cognitive performance.

Accordingly, we have defined cognitive resilience as patients with high cognitive performance after a long-disease duration.

Comment 2. Results: Page 6, end of the first paragraph: As shown in Table 1, there was also no significant difference in visual functions. This fact needs to be included in the narrative as well.

Response: We added to the narrative the fact that there was no significant difference in visual function, Results end of first paragraph.. 

Comment 3. Figure 1 is exactly the same information as in Table 2. Either one, plus the narrative should be sufficient.

Response: We omitted Table 2 and added the data to the narrative. Accordingly we changed the order of the figures (Figure 1is now Figure 2, and Figure 2 is now Figure 1), and explained in the narrative the numbers of the inconsistency groups.

Comment 4. Discussion: Page 7, end of the first paragraph: Can you discuss some of side effects of DMDs?

Added to the Discussion. 

Comment 5. Page 9: beginning of the second paragraph: If this is a prospective study, how is it possible that data was collected prospectively?

We mentioned that the study was retrospective cross-sectional, but the data was collected prospectively – as we wanted to highlight the point that patients were evaluated on an on-going basis prospectively. As it seems confusing we omitted the term prospectively. 

. 

I hope you will find the revised version merit for publication in PlosOne.

With respects

Anat Achiron, MD, PhD

Sheba Medical Center 

Tel-Hashomer, 52621, Israel.

Tel: 972-3-5303932; Fax: 972-3-5348186

Email: Anat.Achiron@sheba.health.gov.il

---

## [Decision Letter · Decision Letter 1]

8 Aug 2019

PONE-D-19-14839R1

Cognitive function in multiple sclerosis: A long-term look on the bright side

PLOS ONE

Dear Prof ACHIRON,

Thank you for submitting your manuscript to PLOS ONE. After careful consideration, we feel that it has merit but does not fully meet PLOS ONE’s publication criteria as it currently stands. Therefore, we invite you to submit a revised version of the manuscript that addresses the points raised during the review process.

We would appreciate receiving your revised manuscript by Sep 22 2019 11:59PM. To enhance the reproducibility of your results, we recommend that if applicable you deposit your laboratory protocols in protocols.io, where a protocol can be assigned its own identifier (DOI) such that it can be cited independently in the future. For instructions see: http://journals.plos.org/plosone/s/submission-guidelines#loc-laboratory-protocols

We look forward to receiving your revised manuscript.

Kind regards,

Abiodun E. Akinwuntan, PhD, MPH, MBA

Academic Editor

PLOS ONE

Additional Editor Comments (if provided):

Good job responding to the comments. Consistency in terminologies has made the manuscript easier to read and understand.

The few corrections needed are:

Page 6: Consider adding "disability" in the sentence with: Neurological disability by EDSS score and the functional system "disability" scores (except for visual functional score) were …..

Page 6, last paragraph: Table 2 shows 45/1010 while the narrative reports 45/1010 for high cognitive performance and severe disability. Which is correct?

Page 7, second paragraph in "Discussion": Consider changing the sentence to "The use of DMD significantly changed the pattern of MS progression."

Overall, check all citations and references for accuracy.

Reviewers' comments:

Reviewer's Responses to Questions

**Comments to the Author**

1. If the authors have adequately addressed your comments raised in a previous round of review and you feel that this manuscript is now acceptable for publication, you may indicate that here to bypass the “Comments to the Author” section, enter your conflict of interest statement in the “Confidential to Editor” section, and submit your "Accept" recommendation.

Reviewer #1: All comments have been addressed

2. Is the manuscript technically sound, and do the data support the conclusions?

Reviewer #1: Yes

3. Has the statistical analysis been performed appropriately and rigorously? 

Reviewer #1: Yes

4. Have the authors made all data underlying the findings in their manuscript fully available?

Reviewer #1: Yes

5. Is the manuscript presented in an intelligible fashion and written in standard English?

Reviewer #1: Yes

6. Review Comments to the Author

Reviewer #1: Thank you for the opportunity to evaluate the manuscript PONE-D-19-14839R1 “Cognitive function in multiple sclerosis: a long-term look on the bright side”. I was not involved in the first review round and will therefore critique the manuscript from a fresh perspective.

This retrospective study includes a large number (n = 1010) of patients with relapsing-remitting multiple sclerosis (MS) and secondary progressive MS. The main aim was to describe cognitive functioning in those patients who had received at least 10 years of disease modifying drug treatment and to correlate cognitive impairments with long term disability. The authors found that the majority of patients with MS did not have cognitive decline. The authors discuss the findings in light of cognitive resilience and long-term quality of life.

Overall, I enjoyed reading the manuscript. The manuscript is topical, timely, and well-written and contributes to the body of knowledge. My major comments are:

1. The discussion around cognitive resilience could be strengthened. Devote at least a paragraph explaining what cognitive resilience is, how it can be captured, what predisposing factors contribute to cognitive resilience, and how that may impact disability and quality of life. Provide references.

2. The leap from good performance on cognitive tests to “cognitive resilience” needs better argumentation. The authors report that 33.7% are cognitive resilient because they score well on cognitive tests despite long disease duration. However, they also have mild disability. It is logical to assume that those with mild disability will also have less cognitive impairments. The 33.7% with good performance in cognitive tests reported here do well because they are generally less disabled; not necessarily because they show better cognitive resilience. You could also argue then that these individuals have better physical resilience, or respond better to disease modifying treatment. It would be more important to focus on the persons who show high cognitive performance despite moderate to severe disability, e.g., those individuals reported in Table 2.

I have few minor comments:

3. How were the three groups of cognitive performance and the three groups of neurological disability determined. Please provide references on the categorization of cognitive performance and neurological disability.

4. Please add in the limitations section that this study included only relapsing-remitting and secondary progressive MS.

5. Please elaborate on exclusion criterion 5. How was it determined from chart review whether a patient had severe upper limb or visual impairment?

6. On that note, visual performance did not differ between the three groups (Table 1), probably because visual impairment was an exclusion criterion. Please add this to the discussion.

7. PLOS authors have the option to publish the peer review history of their article (what does this mean?). If published, this will include your full peer review and any attached files.

Reviewer #1: Yes: Hannes Devos

---

## [Author Response · Author response to Decision Letter 1]

10 Aug 2019

Comment 1. Page 6: Consider adding "disability" in the sentence with: Neurological disability by EDSS score and the functional system "disability" scores (except for visual functional score) were …..

Response: Corrected

Comment 2. Page 6, last paragraph: Table 2 shows 45/1010 while the narrative reports 45/1010 for high cognitive performance and severe disability. Which is correct?

Response: The title column in Table 2 shows: High cognitive performance & Severe disability, so essentially both are the same and correct. 

Comment 3. Page 7, second paragraph in "Discussion": Consider changing the sentence to "The use of DMD significantly changed the pattern of MS progression."

Response: corrected.

Comment 4. check all citations and references for accuracy.

Response: We checked all citations and references to ensure accuracy.

Review Comments to the Author

1. The discussion around cognitive resilience could be strengthened. Devote at least a paragraph explaining what cognitive resilience is, how it can be captured, what predisposing factors contribute to cognitive resilience, and how that may impact disability and quality of life. Provide references.

Response: We have added to the Discussion a paragraph explaining what cognitive resilience is, how it can be captured, what predisposing factors contribute to cognitive resilience, and how that may impact disability and quality of life. In accordance we have provide references as follows:

The term resilience was derived from the Latin words salire (to leap or jump), and resilire (to spring back). When applied to cognition it stands for the capacity of the brain to resist deteriorating processes or injuries (SternY. What is cognitive reserve? Theory and research applications of the reserve concept. J Int Neuropsychol Soc 2002; 8:448–460. McEwen BS. In pursuit of resilience: stress, epigenetics, and brain plasticity. Ann N Y Acad Sci. 2016;1373:56-64). Cognitive resilience literature has focused on specific contexts in which individuals differ in their capabilities to withstand or overcome brain insults and to explain the difference in the patterns of cognitive decline associated with aging and neurodegenerative diseases (de Frias CM, Dixon RA, Bäckman L. Use of memory compensation strategies is related to psychosocial and health indicators. J Gerontol B Psychol Sci Soc Sci. 2003;58:12-22.). Various predisposing interacting factors may contribute to the road map for brain resilience including education, gender, prior brain injuries, family history, participation in cognitively stimulating activities, physical exercises, social relationships and apoE genotype. (Christensen H1, Hofer SM, Mackinnon AJ, Korten AE, Jorm AF, Henderson AS. Age is no kinder to the better educated: absence of an association investigated using latent growth techniques in a community sample. Psychol Med. 2001;31:15-28. Kirk-Sanchez NJ, McGough EL. Physical exercise and cognitive performance in the elderly: current perspectives. Clin Interv Aging. 2014;9:51-62. Hofer SM, Christensen H, Mackinnon AJ, Korten AE, Jorm AF, Henderson AS, Easteal S. Change in cognitive functioning associated with apoE genotype in a community sample of older adults. Psychol Aging. 2002;17:194-208. Wilson RS, Mendes De Leon CF, Barnes LL, Schneider JA, Bienias JL, Evans DA, Bennett DA. Participation in cognitively stimulating activities and risk of incident Alzheimer disease. JAMA. 2002;287:742-8. Seeman TE, Lusignolo TM, Albert M, Berkman L. Social relationships, social support, and patterns of cognitive aging in healthy, high-functioning older adults: MacArthur studies of successful aging. Health Psychol. 2001;20:243-55.). In young patients with a chronic long-lasting disease like MS, characterizing differences between cognitive subgroups in relation to clinical variables may afford new insights into the on-going neuroplasticity mechanisms and in accordance suggest plasticity‐facilitating treatments to enhance cognitive resilience. 

2. The leap from good performance on cognitive tests to “cognitive resilience” needs better argumentation. The authors report that 33.7% are cognitive resilient because they score well on cognitive tests despite long disease duration. However, they also have mild disability. It is logical to assume that those with mild disability will also have less cognitive impairments. The 33.7% with good performance in cognitive tests reported here do well because they are generally less disabled; not necessarily because they show better cognitive resilience. You could also argue then that these individuals have better physical resilience, or respond better to disease modifying treatment. It would be more important to focus on the persons who show high cognitive performance despite moderate to severe disability, e.g., those individuals reported in Table 2.

Response: As the Reviewer commented, indeed, cognitive resilience in patients with long disease duration and mild disability probably signifies MS resilience, e.g., resilience to the disease pathogenic mechanisms. These patients are defined as patients with benign disease course and therefore also manifest with less cognitive impairments. Our findings implicate that cognition as a part of the functional neurological spectrum of MS is better reserved in patients with less active disease, as patients with low disability were found to be more cognitively resilient. 

Regarding the group of patients who show high cognitive performance despite moderate to severe disability as compared to patients with low cognitive performance despite mild disability, we have specified in the Discussion a possible explanation related to anatomical involvement. We suggest that patients with moderate to severe disability but high cognitive performance the spinal cord will be more involved as compared to patients with mild disability but with low cognitive performance that probably present widespread brain disease and relatively lower spinal cord involvement.

We have added these paragraphs to the Discussion.

minor comments:

3. How were the three groups of cognitive performance and the three groups of neurological disability determined. Please provide references on the categorization of cognitive performance and neurological disability.

Response: As specified in the Methods - Cognitive assessment, last paragraph, the three groups of cognitive performance and the three groups of neurological disability were determined according to the global cognitive score (GCS) and the Expanded Disability Status Scale (EDSS) score as follows:

Cognitive performance was categorized by GCS as “high” (GCS >100), “moderate” (GCS 85-100) or “low” (GCS<85). Neurological disability was classified according to EDSS score as “mild” (EDSS ≤3), "moderate” (EDSS 3.5-5.5), or "severe" (EDSS >6.0).

Both are referenced.

We better detailed this by adding the title of Group categorization to this paragraph.

4. Please add in the limitations section that this study included only relapsing-remitting and secondary progressive MS.

Response: Added.

5. Please elaborate on exclusion criterion 5. How was it determined from chart review whether a patient had severe upper limb or visual impairment?

Response: This exclusion criteria is determined as a part of the cognitive test. Patients that have severe upper limb dysfunction like paralysis or tremor are not able to hold the computer-mouse and therefore are technically excluded from preforming the test. Similarly, at the beginning of the cognitive test visual acuity is assessed and patients that can not read the instructions are technically excluded from performing the test. 

The explanation was added to exclusion criterion 5.

6. On that note, visual performance did not differ between the three groups (Table 1), probably because visual impairment was an exclusion criterion. Please add this to the discussion.

 Response: Added.

---

## [Decision Letter · Decision Letter 2]

15 Aug 2019

Cognitive function in multiple sclerosis: A long-term look on the bright side

PONE-D-19-14839R2

Dear Dr. ACHIRON,

We are pleased to inform you that your manuscript has been judged scientifically suitable for publication and will be formally accepted for publication once it complies with all outstanding technical requirements.

With kind regards,

Abiodun E. Akinwuntan, PhD, MPH, MBA

Academic Editor

PLOS ONE

Additional Editor Comments (optional):

Reviewers' comments:

Reviewer's Responses to Questions

**Comments to the Author**

1. If the authors have adequately addressed your comments raised in a previous round of review and you feel that this manuscript is now acceptable for publication, you may indicate that here to bypass the “Comments to the Author” section, enter your conflict of interest statement in the “Confidential to Editor” section, and submit your "Accept" recommendation.

Reviewer #1: All comments have been addressed

2. Is the manuscript technically sound, and do the data support the conclusions?

Reviewer #1: Yes

3. Has the statistical analysis been performed appropriately and rigorously? 

Reviewer #1: Yes

4. Have the authors made all data underlying the findings in their manuscript fully available?

Reviewer #1: Yes

5. Is the manuscript presented in an intelligible fashion and written in standard English?

Reviewer #1: Yes

6. Review Comments to the Author

Reviewer #1: The authors have adequately addressed all comments and I recommend moving forward with publication of this manuscript.

7. PLOS authors have the option to publish the peer review history of their article (what does this mean?). If published, this will include your full peer review and any attached files.

Reviewer #1: Yes: Hannes Devos

---

## [Editor Report · Acceptance letter]

21 Aug 2019

PONE-D-19-14839R2 

Cognitive function in multiple sclerosis: A long-term look on the bright side 

Dear Dr. ACHIRON:

I am pleased to inform you that your manuscript has been deemed suitable for publication in PLOS ONE. Congratulations! Your manuscript is now with our production department. 

With kind regards,

on behalf of

Dr. Abiodun E. Akinwuntan 

Academic Editor

PLOS ONE